# A Pilot-Scale Evaluation of Duckweed Cultivation for Pig Manure Treatment and Feed Production

**DOI:** 10.3390/plants14172680

**Published:** 2025-08-27

**Authors:** Marie Lambert, Reindert Devlamynck, Marcella Fernandes de Souza, Pieter Vermeir, Katleen Raes, Mia Eeckhout, Erik Meers

**Affiliations:** 1Provincial Research and Advice Centre for Agriculture and Horticulture (Inagro vzw), Ieperseweg 87, 8800 Roeselare, Belgium; reindert.devlamynck@inagro.be; 2Lab for Bioresource Recovery, Department of Green Chemistry and Technology, Faculty of Bioscience Engineering, Ghent University, Coupure Links 653, 9000 Ghent, Belgium; marcella.fernandesdesouza@ugent.be (M.F.d.S.); erik.meers@ugent.be (E.M.); 3Laboratory for Chemical Analysis (LCA), Department of Green Chemistry and Technology, Faculty of Bioscience Engineering, Ghent University, Valentin Vaerwyckweg 1, 9000 Ghent, Belgium; pieter.vermeir@ugent.be; 4Research Unit VEG-i-TEC, Department of Food Technology, Safety and Health, Campus Kortrijk, Ghent University, St-Martems Latemlaan 2B, 8500 Kortrijk, Belgium; katleen.raes@ugent.be; 5Research Unit of Cereal and Feed Technology, Department of Food Technology, Safety and Health, Faculty of Bioscience Engineering, Ghent University, 9000 Ghent, Belgium; mia.eeckhout@ugent.be

**Keywords:** duckweed, nutrient removal, novel protein, pilot study, *Lemna*

## Abstract

Livestock-intensive regions in Europe face dual challenges: nutrient surpluses and a high dependency on import of high-protein feedstocks. This study proposes duckweed (Lemnaceae) as a potential solution by recovering nutrients from manure-derived waste streams while producing protein-rich biomass. This study evaluated the performance of duckweed treatment systems at a pig manure processing facility in Belgium. Three outdoor systems were monitored over a full growing season under temperate climate conditions. Duckweed cultivated on constructed wetland effluent showed die-off and low protein content, while systems supplied with diluted liquid fraction and nitrification–denitrification effluent achieved consistent growth, yielding 8 tonnes of dry biomass/ha/year and 2.8 tonnes of protein/ha/year. Average removal rates were 1.2 g N/m^2^/day and 0.13 g P/m^2^/day. Growth ceased after approximately 100–120 days, likely due to rising pH and electrical conductivity, suggesting ammonia toxicity and salt stress. Harvested duckweed had a high protein content and a total amino acid profile suitable for broilers, though potentially limiting in histidine and methionine for pigs or cattle. Additionally, promising energy and protein values for ruminants were measured. Although high ash and fibre contents may limit use in monogastric animals, duckweed remains suitable as part of a balanced feed. Its broad mineral profile further supports its use as a circular, locally sourced feed supplement.

## 1. Introduction

Livestock-intensive regions in Europe face dual challenges, i.e., nutrient surpluses and a shortage of high-protein feedstocks. Duckweed cultivation offers a promising solution to address both issues. Studies have shown that duckweed has good growth on pig manure wastewaters (e.g., the liquid fraction obtained after manure separation and/or the effluent obtained after nitrification–denitrification or digestion), while providing a protein-rich biomass [1,2,3,4]. By utilizing duckweed treatment systems, nutrients from various agricultural waste streams can be efficiently recovered into a protein-rich feedstock. However, research on long-term, outdoor duckweed cultivation systems for sustainable nutrient recovery and feed production remains limited.

Few pilot-scale studies have investigated duckweed-based treatment systems, primarily under tropical or Mediterranean conditions. In Brazil, two ponds fed with biodigester effluent were monitored for one year, showing substantial nitrogen (N) and phosphorus (P) removal and promising biomass and protein yields [1]. Likewise, the LIFE LEMNA project in Spain evaluated duckweed grown year-round on the liquid fraction of pig manure after anaerobic digestion in an outdoor system, focusing on nutrient uptake and its use as a feed or fertilizer [5]. These studies confirm the potential of duckweed in warm climates, but their findings may not be directly transferable to temperate regions.

More relevant to temperate climates, Devlamynck et al. [3,4] evaluated duckweed cultivation on a mixture of liquid fraction and nitrification–denitrification effluent from pig manure during one Belgian growing season. That study assessed macronutrient uptake, biomass productivity, and provided an initial amino acid profile. However, several essential amino acids (such as tryptophan, cysteine, and proline) were either not quantified or fell below detection limits. Moreover, other critical parameters for environmental and agronomic assessment such as digestibility, energy value, organic matter removal (COD, BOD), total suspended solids (TSSs), and sedimentation were not included.

The present study builds on this prior research and expands it with a more comprehensive evaluation of duckweed treatment systems under temperate maritime conditions. Particular focus is placed on assessing duckweed’s value as a feed ingredient, including a full essential amino acid profile, digestibility and energy content evaluation. Environmental performance is evaluated in greater detail, with attention to salinity dynamics, nutrient removal efficiency, including COD, BOD, TSS, and sediment accumulation.

As in the study by Devlamynck et al., two of the three systems in this experiment were supplied with a mix of liquid fraction and nitrification–denitrification effluent [3,4]. In the third system, the use of constructed wetland effluent as a duckweed growth medium was investigated. Constructed wetlands are increasingly used as sustainable wastewater treatment systems, leveraging natural processes to remove pollutants [6]. However, recent performance analyses in Flanders show that despite effective nutrient removal, many wetland effluents still contain nutrient concentrations high enough to pose a risk of eutrophication [6]. This experiment could highlight the potential of duckweed systems as an additional polishing step for effluent treatment.

In summary, three pilot-scale (3 × 10 m^2^) duckweed systems were monitored throughout a full Belgian growing season. This study contributes novel data on feed quality, environmental performance, and system limitations, providing new insights into the long-term viability, challenges, and optimisation strategies for duckweed-based nutrient recovery and circular feed production in temperate agricultural regions. The main objective of this study was to evaluate the dual functionality of duckweed systems for nutrient recovery from manure-derived effluents and feed production under real-life, temperate field conditions.

## 2. Materials and Methods

### 2.1. Pilot Set-Up

Three duckweed treatment systems (each 10.18 m^2^, 0.5 m depth) were set up at a pig farm in Pittem, Flanders (Belgium; 50°59′21.63″ N, 3°17′44.72″ E). Systems 1 and 2 (S1, S2) were filled with the same composed medium, containing streams obtained from the manure treatment process located at the farm. System 3 (S3) was filled with the effluent from the constructed wetland located at the farm.

A schematic overview of the manure treatment process and the streams composing the growing medium of the systems is presented in Figure 1. At the pig farm, the manure treatment process starts with the separation of manure into a solid and a liquid fraction (resp. SF, LF) using a centrifuge. The SF typically contains more P, and the LF is richer in N. After centrifugal separation, the LF is further treated biologically, with nitrification–denitrification (NDN), during which ammonia is converted into nitrogen gas (N_2_). Eventually, an NDN effluent (NDNE) is obtained. In order to further reduce the nutrient content in the NDNE, the effluent is treated in a constructed wetland (CW) until a dischargeable effluent is obtained, which we refer to constructed wetland effluent (CWE) [7].

To determine the start medium composition of S1 and S2, a non-linear solver technique was performed using Microsoft Excel [4,8]. This was calculated with the following constrictions:All fractions of LF, NDNE, and water are greater than zero;The sum of all fractions of LF, NDNE, and water equals 100%;The total N and P contents of the final mixture should be below the limits proposed for *Lemna minor* to obtain optimal growth [9];The N/P ratio of the medium should equal 7.4, as this is the ratio between the N removal and P removal that was determined in an outdoor duckweed system with diluted NDNE as the growing medium [4].

To perform this calculation, the N and P concentrations of the NDNE and LF were first determined. The results of the non-linear solver technique, and thus the composition of the growing medium at the start of the growing season in S1 and S2, are given in Table 1. The N and P concentrations of the water used were negligibly low and were therefore not taken into account.

To prepare the start medium, S1 and S2 were filled with 4355 L water, 571 L NDNE and 74 L LF, following the calculations as presented in Table 1. During the growing season, the systems were fertilised weekly with a ‘fertiliser mix’ that was prepared at the farm every five to six weeks. This mix was stored next to each system in an IBC container (Figure 2a) and was made by mixing 10% LF with 90% NDNE. The IBC containers were covered with a black foil and were mixed every time before a volume was added to fertilise the system.

To fertilise every week in such a way that the N and P inputs would approximately match the expected system removal (including other mechanisms beyond duckweed uptake, like volatilization and sedimentation), the daily nutrient requirements were estimated based on removal capacities reported by Devlamynck et al. [4], who used comparable pig manure-derived effluents (LF and NDNE) in an outdoor set-up under similar climate conditions. Their reported values (1107 ± 715 mg N/m^2^/day, 149 ± 150 mg P/m^2^/day) were used to calculate the weekly nutrient demand of 11.3 g N and 1.5 g P per system. The fertiliser mix (10% LF + 90% NDNE) contained 598.9 mg N/L and 83.8 mg P/L, resulting in a weekly addition of 130 L per system.

In the third duckweed treatment system (Figure 2b), duckweed was grown on the effluent of the constructed wetland (CWE). The N and P concentrations measured in the effluent before installation of the system were, respectively, 29.9 mg/L and 0.53 mg/L. In order to obtain the same N and P uptake as the N and P addition, the system would have to be refilled every 1.8 days for P and every 13.5 days for N. If we calculate further for N, this means that every week, 2638 L of medium should be discharged from the system, and the same amount should be added to the system from the effluent from the wetland. Calculated for P, the weekly discharged and added volume should be 20,033 L. As these volumes are very different and not practically workable, it was decided to weekly discharge a feasible volume (1985 L) of medium and re-add the same amount from the wetland effluent pond. This was practically carried out by discharging water from the system until the water surface was lowered 19.5 cm, followed by refilling the system with effluent from the wetland using a pump.

From May until October 2023, for the three systems, the growing medium was weekly fertilized, along with weekly harvesting. Duckweed was first grown under lab conditions and inoculated on 2 May 2023 on all the systems. Every week, duckweed was harvested, and samples were taken from the growing medium. More details on the harvesting procedure are given in Section 2.4.

On each harvest and sampling day, the total medium volume of S1 and S2 was first standardized by either adding water or discharging growth medium into the constructed wetland until the water level was restored to 0.5 m. After homogenizing the medium by stirring, a first sample of the growth medium was then collected. Next, the IBC containers with the fertilizer mix were manually mixed, and both systems were fertilized. Following fertilization, the growth medium was manually homogenized by stirring, and a part of the duckweed was harvested. Finally, a second sample of the growth medium was taken.

### 2.2. Duckweed Identification

Duckweed grown in the three systems was *L.* ×*japonica* Clone 6580, obtained as *L. minor* but later revealed to be *L.* ×*japonica* [10]. Plant identification was carried out by DNA extraction using the OmniPrep™ Genomic DNA Isolation kit (G-biosciences, Saint Louis, MO, USA). The *atpF-atpH* and *psbK-psbI* noncoding chloroplast spacers were PCR amplified following the methodology described elsewhere [11], and sent for Sanger sequencing at Eurofins Genomics (Ebersberg, Germany) using forward and reverse primers. The obtained sequences were compared with the chromatograms and were corrected when necessary. The sequences from forward and reverse primers were merged into a single sequence, which was blasted against the NCBI genbank.

### 2.3. Analytical Methods

#### 2.3.1. Plant FW and DM Determination

Harvested fresh plant material was drip-dried in a fine mesh fishing net for some minutes prior to determining the fresh weight (FW). With each harvest, a plastic perforated ‘dry bag’ was filled with 300 to 400 g of fresh duckweed that was dried the same day for a minimum of 4 days at 60 °C. Before drying, the full bag was weighed using a digital hanging scale. The dried weight of the duckweed was determined using a bench scale (LA 320P, Sartorius Lab Instruments, Göttingen, Germany) and used to calculate the specific dry matter (DM) content of each harvest.

Prior to further analysis, the moisture content of the stored oven-dried samples was re-evaluated and found to average 8% due to moisture reabsorption during storage. This value was used to correct all concentrations measured on a per-mass basis of dried duckweed, ensuring results were consistently expressed on a 100% dry matter basis.

#### 2.3.2. Compositional Analysis of Duckweed

The total N content (T-N) of the dried duckweed was determined before and after cultivation using a CN analyser (Primacs SNC-100, Skalar, Breda, The Netherlands). Kjeldahl nitrogen (ammonium and organic N; Kj-N) was measured using a digestion unit (525-355/01, ELE International, Milton Keynes, UK) and acid scrubber, distiller (Vapodest 300, Gerhardt, Königswinter, Germany), and titrator (compact Eco Titrator, Metrohm, Herisau, Switzerland).

The crude protein content was calculated as 6.25 times the Kjeldahl total nitrogen content of dried duckweed [12,13,14]. This method was also compared with the protein content of duckweed obtained by directly taking the sum of individual AAs.

For plant Ca, Mg, Na, K, T-P, S, Al, Cu, Fe, Mn, and Zn content, dried plant material, to which 5 mL of 65% HNO_3_ was added, was first digested in a microwave (Milstone Ultrawave SRC technology, Milestone, Sorisole, Italy). Next, samples were accordingly diluted prior to elemental determination with inductively coupled plasma-optical emission spectrometry (ICP-OES) (Qtegra iCAP 7000 Plus, ThermoFisher scientific, Waltham, MA, USA).

All concentrations measured in the duckweed samples were expressed in mg/kg DM.

#### 2.3.3. Analysis of NDNE, LF, Fertiliser Mix and Growing Media

For the determination of the T-N content in liquid samples, a similar procedure as for the determination of Kj-N was followed. The main difference lies in the digestion step, where next to H_2_SO_4_, salicylic acid and Na_2_SO_3_ were used, in accordance with Van Ranst et al.’s study [15].

For the determination of Ca, Mg, Na, K, T-P, S, Cu, Fe, Mn, and Zn concentrations in the waste streams and the more diluted growth media before and after cultivation, the samples were first digested on a hot plate with HNO_3_ (65%) for 30 min without prior filtration. In total, 5 mL of HNO_3_ solution was added gradually during the digestion process until a transparent sample was obtained. After digestion, the sample was filtered through a Whatman, grade 5 filter paper and diluted with Milli-Q water until a total volume of 50 mL was reached. Next, samples were accordingly diluted prior to elemental determination with ICP-OES (Vista-MPX, Varian Inc., Palo Alto, CA, USA).

The determination of Cl^−^, NO_3_^−^, PO_4_^3−^, SO_4_^2−^ in the water samples was assessed using ion chromatography (761 Compact Ion Chromatograph, Methrom, Herisau, Switzerland), preceded by 0.45 µm syringe filtration and dilution. All concentrations measured in water samples or effluents (NDNE, LF, CWE) were expressed in mg/L.

pH and electric conductivity (EC) were measured in situ with a portable probe (HI991301, Hanna Instruments, Padova, Italy) or afterwards in the lab with a pH-meter (ProfiLine pH 3110, WTW, Weilheim, Germany) and a conductivity tester, respectively (ProfiLine Cond 3110, WTW, Weilheim, Germany).

For the evolution of the chemical oxygen demand (COD), biological oxygen demand (BOD), and total suspended solids (TSSs), system 1 was sampled every week. The sampling was conducted alternately before and after fertilizing; more specifically, in week ‘x’, a sample was taken before fertilizing, and in week ‘x + 1’, a sample was taken after fertilizing. The samples were always analysed within 24 h after sampling and stored in the fridge at 2–5 °C prior to analysing. First, the COD was measured according to ISO 6060 [16], followed by the determination of the BOD, according to ISO 5815-1 [17]. The TSS was determined by filtering the sample through a 0.45 µm membrane after homogenization and drying until constant weight is achieved.

The sedimentation degree, or the measurement of settleable solids, was assessed using an Imhoff Cone (Brand, Wertheim, Germany). At the end of the monitoring season (19/10), S1 and S2 were thoroughly homogenized, and a 2 L sample was taken. Within 24 h, this sample was brought to the lab, again homogenized, and poured into the Imhoff cone to the 1 L mark at room temperature. After 45 min, the cone was gently rotated clockwise and counterclockwise to release the suspended matter clinging to the sides of the Imhoff cone. After another 15 min, the volume of settleable solids in the cone was recorded (mL/L).

#### 2.3.4. Nutritional Analyses for Feed Value Determination

##### Sample Preparation

Four duckweed samples were prepared and analysed, which will be referred to as Mixes 1, 2, 3 and 4. Mix 1, 2, and 3 consist of a mixture of dried duckweed samples harvested from S1 and S2 at the beginning (11/05–22/06), the middle (29/06–10/08) and at the end (17/08–28/09) of the growing season, respectively. The distribution of the specific proportion of each sample in the mixtures is provided in Appendix B, Table A1. Due to an oven malfunction during drying, the sample from the 03/08 harvest was lost and therefore not included in Mix 2. Mix 4 is composed of an equal share of Mix 1, Mix 2, and Mix 3.

##### Amino Acid (AA) Analysis

The AA composition of Mix 4 was quantified in duplicate using reversed-phase high-performance liquid chromatography (RP-HPLC) with absorbance detection by a diode array detector (DAD).

Prior to chromatographic analysis, samples underwent hydrolysis to release the AAs from their proteinaceous matrix. For most of the determined AAs, acid hydrolysis was performed prior to RP-HPLC analysis. Cyst(e)ine and methionine were first oxidized to cysteic acid and methionine sulphone, respectively, prior to acid hydrolysis. For the determination of tryptophan, the samples were hydrolysed under alkaline conditions. The hydrolysis methods were based on ISO 13903:2005 [18] and AOAC Method 988.15 [19].

After hydrolysis, the samples were filtered through a 0.45 μm syringe filter (25 mm, PTFE, VWR, Leuven, Belgium) and transferred into 2 mL glass vials with slitted screw caps for HPLC analysis. AA analysis was performed applying the standard operating procedure (SOP) of Agilent Technologies (Santa Clara, CA, USA) using ortho-phthalaldehyde (OPA)/9-fluorenylmethyl chloroformate (FMOC) online derivatization in an Agilent 1290 Infinity II LC system. The AAs were first converted into OPA and FMOC derivatives using the 1260 Infinity II Vialsampler (Agilent, Santa Clara, CA, USA), after which separation was performed on an InfinityLab Poroshell 120 HPH-C18 column (4.6 mm × 100 mm × 2.7 μm; Agilent, USA). The mobile phases, at a flow rate of 2 mL/min, consisted of 10 mM Na_2_HPO_4_, 10 mM Na_2_B_4_O_7_, 0.5 mM NaN_3_ at pH 8.2 (eluent A) and acetonitrile/methanol/mQ in a ratio of 45/45/10 (*v*/*v*/*v*) (eluent B), and followed the gradient in the SOP. The absorbance was measured at 262 nm for FMOC-amino acids (Pro and Hyp) and at 338 nm for OPA-amino acids (others). Calibration FAA standard solutions ranging from 22.5 to 900 μM and internal standards (0.5 mM) were used for quantification. Norvaline was used as internal standard for the OPA-amino acids, and sarcosine for the FMOC-amino acids.

To evaluate the protein quality, the AA composition was compared to what was found in earlier research and to the nutritional requirements of humans and broilers found in the literature [20,21]. An essential amino acid index (*EAAI*) was calculated using the following equation [22]:(1)$$EAAI=\sqrt[{n}]{\frac{aa1}{AA1}*\frac{aa2}{AA2}*\ldots *\frac{aan}{AAn}}$$
with *aa*1, *aa*2, …, *aan* representing the percentage of the respective EAA content in the sample and *AA*1, *AA*2, …, *AA*n representing the required levels of the respected EAAs of the specific species. For the calculation of the EAAI for humans, the FAO/WHO UNU 2007 Reference Pattern for adults (>18 years) was used [16]. For broiler chickens, the amino acid requirements for chicks from 0 to 21 days of age were used as a reference [21]. In addition to the nine essential amino acids (EAAs) considered in the *EAAI* calculation for humans, arginine was also included in the assessment for broilers. This is due to their limited capacity to synthesize Arg on their own.

##### Proximate Analysis

Different nutritional parameters were determined for Mixes 1, 2, 3 (crude protein, crude fat, ash), and 4 (rest) to determine the feed value. A summary of the analysed parameters and their corresponding methods and calculations are given in Table 2.

The energy value for ruminants was estimated using the Dutch system and was expressed in VEVI (Feed Unit Growth) and VEM (Feed Unit Lactation) [31]. This was carried out for the four duckweed mixtures (1 mm particle size), specified in Table A1, in order to compare the energy value of duckweed harvested at the beginning, middle, and end of the growing season.

To estimate VEVI-VEM, in vitro rumen incubations were executed by the ILVO research centre in Belgium. Two methods were used in which the difference lay in the inoculum used, being rumen fluid from sheep or the enzyme cellulase. Because it was suspected that there were elements present in duckweed that inhibited microbial growth in vitro, killing off the inoculum and halting organic matter degradation after a certain time, it was chosen to estimate VEVI and VEM with cellulase as the inoculum, as described by De Boever et al. [33]. The digestibility coefficient for organic matter was estimated with the obtained OMDc and the crude ash content.

Next, the duckweed protein value was evaluated through the parameters DVE (Digestible Crude Protein in the Small Intestine) and OEB (Degraded Protein Balance). For this, an in sacco-in vivo rumen incubation was carried out by ILVO. This analysis was performed on the mixed duckweed sample (mix 4 of Table A1).

Duckweed was grinded and sieved to a 3 mm particle size and 2.5 g of DM equivalent was weighed into nylon bags (10 × 8 cm, 37 µm pore size). The rumen incubations were carried out in three lactating cows, which were fed a well-balanced diet of grass and maize silage (~50/50 on DM), supplemented with concentrate feed and (rumen resistant) soybean meal to meet energy and protein requirements. Incubation times were 3, 8, 24, 48, and 336 h, starting just before the morning feeding. To ensure sufficient residue for analysis, 6 bags (2/cow) were incubated for 6 h, 9 bags (3/cow) for 8–24–48 h, and 12 bags (4/cow) for 336 h. For the longest incubation period, the bags contained double the sample quantity. After the incubation period, the bags were immediately submerged in ice water to stop all microbial activity, and adhering particles were rinsed off with tap water. After draining, the bags were stored in a freezer to kill remaining microorganisms. Before further analyses, the bags were thawed and washed for 45 min with cold water in a washing machine (wool program, no spin). Three bags with duckweed were washed without prior rumen incubation (0 h time point) to analyse the washable fraction. After washing and draining, the bags were freeze-dried and weighed, and incubation residues from the three cows were pooled for analysis.

The moisture, ash, crude protein, NDF, and starch content of the residues was determined. Additionally, the (R)NSP was calculated to derive the degradation characteristics of organic matter (OM), crude protein (CP), NDF, starch, and (R)NSP. The protein value (DVE and OEB) could then be derived from these degradation characteristics [32].

### 2.4. Calculations and Statistics

During weekly harvesting, the duckweed mat was never completely removed. A portion was always left in place to ensure full surface coverage in order to suppress algal growth. Because the duckweed surface density before and after harvesting was not measured, the relative growth rate (RGR) could not be calculated, as is commonly performed in duckweed research [4,8,34]. Therefore, absolute productivity was determined by dividing the harvested dry matter (DM) by the system’s surface area (10.18 m^2^) and the number of days since the previous harvest. Since duckweed was cultivated continuously throughout the growing season, total annual biomass and protein production were calculated and reported.

The nutrient removal by each system (referred henceforth as system removal) was calculated weekly using the volume (V) and nutrient concentration (c) of the medium after (a) fertilisation in one week (t) and before (b) fertilisation in the following week (t + 1), according to the following equation:(2)$$System\;nutrient\;removal\left[mg/m^{2}/d\right]=\frac{{\left(V_{a}*c_{a}\right)}_{t}-{\left(V_{b}*c_{b}\right)}_{t+1}}{surface*days},$$
with $V_{a}$ and $V_{b}$ each corrected for the amount of fertilization mix added and medium evaporated/precipitated during that week. The amount of nutrients removed was divided by the surface area of the system (10.18 m^2^) and the number of days since the last fertilisation. A part of the nutrient removal in a duckweed treatment system is due to nutrient uptake from the duckweed that is harvested weekly. This nutrient removal by harvest was calculated as follows:(3)$$Nutrient\;removal\;by\;harvest\left[mg/m^{2}/d\right]=\frac{{\left(DM*dc\right)}_{t}}{surface*days},$$
with the nutrient content of the duckweed (dc) and its dry matter (DM) harvested at one week (t). The calculated removal was normalised to the system’s surface area and the number of days in the fertilisation interval.

Additionally, the percentage of nutrient removal by weekly harvest was calculated as follows:(4)$$Nutrient\;removal\;by\;harvest\left[\%\right]=\frac{(Nutrient\;removal\;by\;harvest{)}_{t}*100}{(Total\;nutrient\;removal{)}_{t}}$$
which was only calculated when a positive system removal was obtained.

Each week, extra nutrients were added to the system as LF and NDNE (S1, S2) or as constructed wetland effluent (S3). The aim was to maintain a constant N and P concentration during the growing season, meaning that the N and P addition should meet the N and P removal in the systems. However, this is not the case for other elements like K, Na, Ca, Mg, Cl or S. The net balance between the addition and removal of certain elements can be positive or negative, leading to either a gradual increase (accumulation) or decrease (depletion) over time. Using linear regression, the accumulation or depletion rate of those elements was determined followed by a Pearson correlation test evaluating its significance level (*p* < 0.05).

Microsoft Excel and R software (R 3.6.1) were used for statistical data processing and visual display. All hypotheses were evaluated on a 5% significance level (*p* < 0.05). The correlation between parameters was determined using the Spearman correlation test. Additionally, a principal component analysis (PCA) was conducted on the set of environmental and chemical variables measured in the growing medium. All variables were standardized prior to analysis to ensure equal weighting. Protein content and productivity were included as supplementary variables, allowing their relationships with the principal components to be visualized without influencing the ordination. Accumulation and depletion rates of pH, EC, and (micro)nutrients in the growing medium were determined using linear regression, and their significance was assessed using a Pearson correlation test.

## 3. Results and Discussion

### 3.1. Agronomic and Environmental Performance

#### 3.1.1. Partially Treated Pig Manure Is a Potential Nutrient Source for Protein-Rich Duckweed Biomass

The duckweed treatment systems S1 and S2 were both fertilised weekly with a mixture of the liquid fraction (LF) and nitrification–denitrification effluent (NDNE) from pig manure processing.

Figure 3 reports the biomass (dry matter) and protein production of these systems over the growing season. The average weekly biomass productivity of both S1 and S2 was 5.5 ± 2.6 g DM/m^2^/day. This yield is consistent with previous research in Flanders using similar nutrient sources, which reported an average biomass productivity of 6.1 ± 2.5 g DM/m^2^/day [3].

Weekly production rates showed some variability, partly due to inconsistencies in the harvesting method. Since only part of the duckweed was harvested each week and the remaining fraction was not precisely controlled, slight differences in post-harvest coverage may have affected the weekly productivities. Nevertheless, a clear trend in duckweed productivity can be observed. As shown in Figure 3, duckweed biomass production showed a steep increase during early summer, peaking between 15/06 and 13/07, followed by a slower decline until complete die-off in late September. The peak productivity during summer is likely explained by favourable weather conditions, particularly high global radiation and long daylight hours (Figure A1).

Duckweed growth ceased on 21/09 in S1 and 28/09 in S2, resulting in growing seasons of 134 and 141 days, respectively, which are shorter than the anticipated 175-day growing season for temperate Belgian conditions. Although an earlier start (e.g., March–April) could have extended the growing period, the early growth cessation is unexpected given that environmental conditions remained suitable throughout September (Figure A1). A longer growing season should have been achievable under these conditions [3]. Therefore, potential causes of this early production halt are further investigated in Section 3.2.

Figure 3 also shows that the protein concentration of the harvested duckweed fluctuated between 30 and 45% throughout the growing season. These values position duckweed as a promising high-quality protein feedstock. However, if duckweed is to be used as a reliable protein source in feed formulations, it is crucial that its protein concentration remains stable over time. To investigate which factors may explain the observed fluctuations in protein content, a Spearman correlation analysis was performed. This analysis revealed a significant negative correlation between protein concentration and total biomass productivity (Spearman’s r = –0.4, *p* = 0.01), suggesting a trade-off between yield and protein concentration. Additionally, the correlation plot (Figure A3) indicated other significant negative associations between protein content and both day duration and iron (Fe) concentration in the medium. However, these relationships are likely indirect, mediated through their influence on productivity. To the best of our knowledge, this inverse relationship between duckweed biomass productivity and protein content has not been previously reported in the literature.

Table 3 extrapolates the productivity and protein content to a larger scale. It is apparent that the duckweed in S3, fertilized with effluent from the constructed wetland, showed very poor to no growth and had a considerable low protein concentration. Without any prior processing of the effluent, this medium is not suitable for duckweed cultivation because of its too high electrical conductivity (EC) (Figure A2).

Duckweed cultivated in systems S1 and S2, fertilized with partially treated pig manure (LF and NDNE), showed good productivity, comparable to previous findings by Devlamynck et al. (Table 3) [3]. Biomass yields were lower than those reported in pilot systems operating under warmer climatic conditions, such as in the Spanish study (17 tonnes DM/ha/year) and the Brazilian study (68 tonnes DM/ha/year) [1,5]. Nonetheless, even within the temperate Belgian climate, the duckweed protein yield obtained in this study suggests strong potential as an alternative to conventional protein sources. For reference, Brazil’s 2023 soybean production averaged 3.4 tonnes/ha with a crude protein content of approximately 35.8% [35,36], corresponding to an average protein yield of about 1.2 tonnes/ha/year. Notably, the annual protein yield achieved in S1 and S2 exceeded this value by a factor of 2.2 and 2.8, respectively. Therefore, considering the significant potential for further optimization of these systems, discussed in Section 3.2, duckweed cultivation in such setups emerges as a promising local alternative to soybean protein production.

#### 3.1.2. Duckweed Growth Resulted in Substantial Nutrient Removal

In addition to its agronomical performance, the environmental performance of the duckweed treatment system was evaluated, focusing specifically on its nutrient removal capacity. In plant-based systems, nutrients are removed via three main processes, which are sedimentation, plant uptake, and microbiological processes like nitrification–denitrification [1,37]. Additionally, dependent on the pH and temperature of the growing medium, nutrient losses can also occur via volatilization, particularly of ammonia.

By weekly sampling of the growing medium, both before and after fertilization, along with nutrient analysis of the harvested duckweed, the overall system removal and the specific removal through duckweed harvesting could be calculated. The removal by plant uptake could not be determined as not all of the duckweed biomass covering the system was harvested every week. The average overall and specific removals for S1 and S2 together are shown in Table 4.

The variation within removal rates, presented in Table 4, is substantial, which may be attributed to the inherent random variation characteristic of biological systems, as has been observed in previous research by Devlamynck et al. [4]. However, negative overall system removal rates were occasionally observed when medium nutrient concentrations measured just before harvesting were higher than those measured after fertilisation one week earlier. This could be attributed to inconsistent mixing of the system prior to sampling or nutrient release from the sediment during mixing before sampling. While these negative values were included in the calculation of overall nutrient removal, they were excluded from the calculation of the specific nutrient removal attributable to duckweed harvest.

Seasonal variation in removal rates was also apparent, but no consistent trends could be observed from the weekly measured nutrient concentrations in the medium. Spearman correlation analysis did not show any significant correlation between nutrient removal and productivity or harvest date. Only pH, T and precipitation had a small positive correlation with the system removal of certain nutrients, as presented in Figure A5.

Despite the high variability, nutrient removal rates observed in S1 and S2 were generally consistent with those reported in previous studies. For nitrogen (N), an average removal rate of 1.2 g/m^2^/day was recorded, closely aligning with previous findings by Devlamynck et al., who reported N removal rates of 1.1 g/m^2^/day under similar wastewater conditions and weather patterns [3,4]. The nitrogen removal attributable to harvest in this study also fell within the range of N uptake previously observed (327 ± 107 and 264 ± 123 mg N/m^2^/day) [3,4]. The proportion of total nitrogen removed through harvesting was higher in this study (48 ± 74%) compared to the previous research (27% and 24% [3,4]).

For phosphorus (P), previous studies reported system removal rates of 0.13 and 0.37 g/m^2^/day, with P uptake rates of 67 ± 26 and 58 ± 31 mg P/m^2^/day and relative uptake percentages between 17% and 39% [3,4]. In the current study, similar system removal (0.13 g/m^2^/day) and uptake rates (72 ± 40 mg/m^2^/d) were achieved, but with a notably higher proportion of phosphorus removed via harvest (97 ± 136%). This suggests that nearly all P removal was attributed to duckweed uptake, in contrast to earlier studies where sedimentation played a more prominent role. A plausible explanation is that phosphorus resuspension from the sediment during water mixing prior to sampling may have elevated the measured concentrations (both before and after fertilization), thereby possibly underestimating the actual system removal. Consequently, the contribution of non-uptake pathways, such as sedimentation, may have been underestimated, resulting in a possible overestimation of the harvest-based share. Moreover, the harvest-based removal percentages shown in Table 4 should not be directly compared to the relative uptake values reported by Devlamynck et al. [3,4], as these were estimated using average concentrations and reflect total uptake rather than removal through harvesting.

Although the harvest-based share of total phosphorus (P) removal may have been overestimated, the opposite pattern was observed for highly soluble elements such as potassium (K) and sodium (Na), which are typically removed primarily through plant uptake. In this study, the relatively low calculated harvest-based removal percentages for K and Na likely resulted from high variability in the weekly system removal estimates. The K removal via harvest in the present study (255 ± 144 mg/m^2^/day) was comparable to previously reported plant uptake rates (233 ± 85 mg/m^2^/day), while the overall K removal was considerably higher and more variable [3,4]. Other elements considerably removed via harvesting included calcium (Ca), manganese (Mn), magnesium (Mg), and sulphur (S), all showing harvest-based removal rates comparable with uptake values reported by Devlamynck et al. [3,4].

Sedimentation is expected to account for a substantial share of nutrient and metal removal in duckweed treatment ponds, particularly for phosphorus and heavy metals. At the end of the monitoring period, sedimentation was assessed in both systems, and only a small amount of settled solids was measured (4.5–5.25 mL/L), corresponding to an estimated sediment depth of only ~0.2–0.3 cm. This limited accumulation is probably due to repeated resuspension of solids during weekly mixing, which hindered stable sediment formation. Future studies should consider operating undisturbed systems to better evaluate the sedimentation dynamics and accurately distinguish between removal pathways.

Mixing also complicated the assessment of BOD, COD, and TSS removal in system 1, where these parameters were monitored. Although some fluctuation was observed between individual measurements, a consistent pattern was observed, particularly during the first half of the growing season. TSS concentrations were generally lower one week after fertilization compared to before (Figure A6c). Similar trends were observed for BOD and COD (Figure A6a,b), suggesting positive removal throughout this period. However, as with most nutrients, the calculated seasonal average removals of BOD, COD, and TSS showed large standard deviations (Table 4). Furthermore, both BOD and COD concentrations increased substantially over time, from 12 mg/L and 418 mg/L O_2_ on May 11th to 108 mg/L and 1612 mg/L O_2_ by September 28th, respectively (Figure A6a,b). The observed low BOD/COD ratio (<0.1) indicates that most of the organic matter was not readily biodegradable, limiting microbial breakdown and explaining the low removal efficiency. In contrast, systems with higher organic loading have demonstrated effective BOD, COD, and TSS removal in duckweed-based treatments [38,39,40], likely due to increased microbial activity supported by the duckweed layer, which enhances oxygen availability and provides surfaces for microbial colonization [38,39].

Compared to other wastewater treatment technologies, such as reed-based constructed wetlands, the monitored duckweed system demonstrated a competitive environmental performance. For instance, Meers et al. [7] reported average nitrogen and phosphorus removal rates of 0.89 g N/m^2^/day and 1.4–2.7 g P/m^2^/day, respectively, in a full-scale constructed wetland treating similar pig manure effluent. While these systems are effective, they typically yield low-nutrient biomass and are harvested infrequently. Duckweed systems, on the other hand, not only facilitate efficient nutrient capture but also frequently generate nutrient-rich biomass suitable for valorisation. Constructed wetlands may, however, provide a valuable complementary post-treatment step, particularly for the removal of organic matter via sedimentation, adsorption, and microbial degradation processes that may be limited in duckweed systems under low BOD/COD conditions.

Additionally, Zimmo et al. compared duckweed and algae-based treatment systems and found that algae systems achieved higher overall nitrogen removal rates [37]. However, under conditions of low organic loading, which are more comparable to our study, duckweed was responsible for a greater proportion of nitrogen assimilation, resulting in higher nitrogen recovery into biomass [37]. For phosphorus, they observed that organic loading had little influence on total phosphorus removal in either system [37]. Interestingly, the highest phosphorus removal occurred in the duckweed-based system [37].

### 3.2. Evaluation of the Pilot Set-Up

#### 3.2.1. Waste Stream Variability Caused an N and P Imbalance

The fertilisation strategy (see Section 2.1) aimed to match weekly N and P additions to estimated system removal capacities, based on values from a previous outdoor duckweed study conducted under similar conditions [4]. However, N and P gradually accumulated over the growing season (Figure 4), suggesting that the actual nutrient loading exceeded removal.

A possible explanation for the imbalance between N and P additions and system removal could be a potential overestimation of the system’s removal capacity in the fertilisation strategy. However, this strategy was based on nutrient removal rates reported by Devlamynck et al., which were highly comparable to those measured in the current study [4]. This suggests that the fertilisation targets were appropriate, and that the observed accumulation resulted primarily from another cause.

This imbalance is most likely explained by the variability in the nutrient content of the applied fertiliser mix, which was based on a single pre-season characterisation of LF and NDNE. These manure-derived waste streams are known to fluctuate considerably in nutrient concentrations over time [4,8]. As the experiment was conducted under semi-practical, on-farm conditions, this approach was intentionally chosen to simplify implementation. Nevertheless, the observed accumulation highlights the importance of periodic monitoring to improve nutrient dosing and avoid excessive nutrient loading when applying this technology.

Although nutrient concentrations in both systems were similar for most of the season, system 2 received one batch with substantially elevated nutrient content, contributing to the stronger accumulation observed there (Table A2).

As further discussed in the following Section 3.2.2, this nutrient build-up may have contributed to the observed decline in duckweed productivity later in the season.

#### 3.2.2. Growth-Limiting Factors: Nutrient Toxicity, Ion Ratios, pH and Salinity

Although duckweed growth was generally good, biomass yields did not reach their full potential, and the growing season ended early despite still favourable climatic conditions (see Figure A1). This indicates that other factors have limited growth. Table 5 compares the measured nutrient concentrations with optimal and maximum growth thresholds from the literature [9,41]. However, these thresholds, which are typically derived from studies using synthetic media, may not directly apply to organic nutrient sources. Previous studies have shown improved duckweed performance in organic fertilization systems even in the presence of “non-optimal” conditions [3,8].

Among all measured elements, only phosphate (PO_4_^3−^) exceeded the maximum concentrations reported by Landolt et al. for optimal duckweed growth [9]. However, additional (unpublished) tests showed that *Lemna* spp. can tolerate concentrations above this threshold without apparent growth inhibition. This suggests that PO_4_^3−^ was not the (only) growth-limiting factor in this system.

Beyond individual elemental toxicity, nutrient interactions can also contribute to growth inhibition. For example, Walsh et al. observed that low Ca:Mg ratios (<1:1.6) in dairy wastewater reduced duckweed growth [42]. This means that die-off occurred when the relative Mg concentration was higher than 1.6. In our study, the relative Mg concentration initially declined but showed an upwards trend till the end of the growing season. Nevertheless, it remained below the critical threshold of 1.6 throughout the growing season. Hence, it is unlikely that the Ca:Mg ratio was a limiting factor (see Figure A7).

Next, pH-dependent toxicity was considered. The toxicity of nitrogen compounds, particularly ammonium (NH_4_^+^), is strongly influenced by pH due to the NH_4_^+^/NH_3_ equilibrium. With increasing pH, a greater proportion of ammoniacal nitrogen exists as NH_3_, which is significantly more toxic. In system 1, NH_4_^+^ concentrations were evaluated against a dynamic toxicity threshold that accounted for pH-dependent NH_3_ formation, following the approach of Devlamynck et al. [3]. While NH_4_^+^ concentrations remained below the adjusted toxicity threshold throughout the season, the increasing pH toward the end of the growth period substantially reduced the safety margin, potentially contributing to suboptimal growth conditions (Figure A8).

Although pH may increase the toxicity of certain elements, it could also decrease the bioavailability of others. However, there was no consistent elemental depletion observed in the weekly harvested biomass (see Figure A9). This suggests that nutrient deficiency was unlikely to be the main growth-limiting factor.

A more plausible explanation for the observed duckweed toxicity is progressive salt stress over the course of the season. Electrical conductivity (EC), which reflects salinity levels, was measured on each harvest day and is shown in Figure A2. Although EC levels remained below the acute toxicity threshold for duckweed [9], a steady increase was observed throughout the season. Previous research showed that such gradual increases in EC, even when remaining below toxic thresholds, can reduce duckweed growth significantly [8].

To further investigate which environmental and chemical factors may have limited duckweed productivity, a Spearman correlation analysis followed by a principal component analysis (PCA) was performed using data from systems 1 and 2 throughout the growing season. As shown in the correlation matrix and PCA biplot (Figure A5 and Figure A6), productivity was significantly positively correlated with day duration, wind speed, solar radiation, and the medium Fe concentration, while significant negative correlations were observed with relative humidity and medium EC, pH, and several dissolved nutrients (Cl^−^, PO_4_^3−^, N, Zn, Co, Mg, Cu, Na, K, P, and S). In contrast, productivity was oriented away from many nutrient variables, which tended to cluster together, reinforcing their collective association with reduced growth. However, it should be noted that many of these dissolved nutrients were also strongly correlated with one another due to shared accumulation patterns over time. As a result, their individual effects on productivity cannot be clearly distinguished, and the observed correlations likely reflect a general stress effect from nutrient buildup rather than the impact of specific elements.

#### 3.2.3. Managing Nutrient Accumulation Through System Design

As shown in Table A3, various nutrients accumulated in the duckweed systems over the course of the growing season. This buildup likely resulted from nutrient inputs exceeding the systems’ removal capacities, indicating an imbalance between loading and treatment. To prevent excessive accumulation, several system design and management strategies can be considered.

One approach is to reduce the loading rate, aligning nutrient inputs with the system’s known removal capacity for each element. Based on measured nutrient removals in S1 and S2, the annual nitrogen treatment capacity amounted to 2153 kg N/ha/year. This implies that, without nitrogen accumulation, a 1 ha duckweed system could theoretically treat 3197 tonnes of the applied fertiliser mixture (2877 tonnes NDNE and 320 tonnes LF). However, under this scenario, phosphorus would exceed its removal capacity and accumulate in the system. To prevent this, dimensioning based on phosphorus removal (216 kg P/ha/year) would reduce the allowable annual loading to approximately 2102 tonnes of fertiliser mix. Although conceptually straightforward, this approach is challenging in practice due to the high temporal variability of manure-derived effluents and the difficulty of accurately determining removal capacities for all nutrients. As previously discussed in Section 3.2.1, while N and P removal were relatively consistent with earlier studies, the removal of other elements such as K, Ca, and PO_4_ showed high variability, likely due to sampling artefacts introduced by weekly mixing.

Additionally, the system’s removal capacity can be enhanced by design modifications. Expanding the duckweed-covered surface increases uptake potential while increasing pond depth enhances dilution (buffering), reducing the toxicity risk from accumulating soluble elements [8]. Moreover, selective pre-treatment of the most limiting nutrients could be applied. In this study, phosphorus was identified as the most limiting element; strategies such as phosphate precipitation or adjusting the LF:NDNE ratio (in favour of NDNE, which contains relatively less phosphorus) could reduce phosphorus loading. Notably, the observed N:P removal ratio was 10, higher than the assumed 7.4, which explains the more rapid accumulation of phosphorus.

### 3.3. Feed Value

#### 3.3.1. Duckweed Is a Valuable Mineral Feed Supplement

Duckweed, when dried, contains not only a high protein concentration but also a diversity of valuable minerals, also referred to as (micro-)nutrients [4]. At each harvest point, samples were dried and analysed for their nutrient composition. Table 6 presents the minimum and maximum concentrations measured in dried duckweed (on 100% DM) from S1 and S2 (also presented as a boxplot in Figure A9). These values are compared to those of soybean meal (100% DM) and the established nutrient requirements for broilers, pigs, and cattle, based on the literature sources [43,44,45,46,47].

Weekly monitoring of the nutrient composition of the harvested duckweed revealed temporal variation in mineral concentrations. Although trends were not always consistent, both S1 and S2 exhibited similar fluctuation patterns. Mg remained relatively stable throughout the growing season, with only minor outliers. Zn, Fe, Cu, and Ca displayed elevated concentrations in the early season, followed by more stable values. Co, Na, and K showed a gradual upward trend, although K concentrations declined again in system 2 later in the season. Mn exhibited a pronounced mid-season peak, whereas P and S fluctuated irregularly with a slight downward trend.

Several of these trends can be linked to changes in nutrient concentrations in the growth medium. For instance, increasing levels of Fe, Co, Na, K, and Mn in the duckweed biomass reflect similar accumulation patterns observed in the medium (Table A3). In contrast, for Mg, although accumulation in the water was observed, no clear increase in plant biomass was detected, suggesting an absence of luxury uptake. A similar lack of proportional enrichment was noted for Zn, Cu, P, and S, indicating that internal regulation mechanisms or uptake thresholds might limit their accumulation in the biomass. Despite a decreasing Ca concentration in the medium over time, no corresponding decline was observed in duckweed Ca content, suggesting homeostatic regulation of essential minerals within the plant tissue. The specific nutrient concentrations of the harvests and the medium samples, varying over the season, can be found in the Appendix A.

Given its high mineral content, the use of dried duckweed as a feed ingredient requires careful consideration to avoid surpassing the maximum tolerable mineral levels established for different livestock species. In this study, the maximum mineral concentrations measured in duckweed (Table 6) exceeded these requirements for several elements, depending on the animal species. Hence, small portions of duckweed can already fulfil the nutrient requirements of feed.

A safety consideration is duckweed’s potential to accumulate unwanted or toxic elements. When grown on heavy metal-rich media, duckweed has the ability to take up heavy metals (i.e., Cd, Pb, As, Hg) to levels above feed safety limits. To assess this, pooled samples from early (Mix 1), mid (Mix 2), and late (Mix 3) stages of the growing season were tested for non-essential heavy metals and metalloids, i.e., Cd, Pb, As, and Hg. As shown in Table 7, all measured concentrations were well below the maximum allowable levels specified by EU Directive 2002/32/EC [49], indicating the safety of duckweed for feed use. These results are consistent with earlier studies, which also reported values within regulatory limits, except for one isolated Pb measurement slightly above 5 mg/kg [4]. Notably, Hg was not analysed in that earlier work but was included in this study and found to be absent in all samples. In the research by Holshof et al., heavy metal levels also remained within regulatory norms [50].

#### 3.3.2. Duckweed Protein Has a Balanced Amino Acid Profile

The amino acid (AA) composition of duckweed was analysed in a pooled sample (Mix 4), revealing a total AA content of 39.9% on DM, consistent with the crude protein value measured using the Kjeldahl method (37.3% DM). Figure 5 presents the essential amino acid (EAA) profile as a percentage of the total protein. It should be noted that only total AA concentrations were measured, and no assessment of their digestibility was carried out. The EAA profile aligns closely with results from previous studies [3] and even exceeds some findings reported in the literature [13,51] (see Table A4).

As shown in Figure 5, duckweed protein contains all EAAs in sufficient quantities to meet the requirements for growth and performance of broilers [21]. Methionine, often the first limiting AA in poultry diets [53], was found in combination with cysteine at higher concentrations than in soybean meal and fishmeal [48,52], resulting in a favourable sulphur amino acid (SAA) profile. Tryptophan, another commonly limiting AA, was also present at higher levels than in conventional protein sources. These results support the potential of duckweed as a complementary or alternative protein source in broiler nutrition. The essential amino acid index (EAAI) for broilers was 1.4, higher than that of soybean meal (1.3) and fishmeal (1.2), indicating superior amino acid balance.

For pigs, AA requirements are typically expressed relative to lysine, which is often the first limiting AA in swine diets [53]. Based on comparisons with recommended AA-to-lysine ratios (Table A5), methionine appeared as the most limiting AA in duckweed protein, particularly relevant for pregnant sows. Histidine was also found to be below or equal to the recommended level for growing pigs and pregnant sows. These findings align with earlier observations by Devlamynck et al., though the histidine deficiency was not observed in that study [3].

For cattle, defining AA requirements is more complex due to rumen microbial protein synthesis, which usually supplies the majority of the crude protein flowing to the small intestine [54]. In general, lysine, methionine, and histidine are frequently discussed as the most limiting amino acids [54]. Therefore, the limitations of duckweed protein for cattle feed are comparable to those discussed for pig feed.

For human nutrition, duckweed also shows strong potential. All EAAs were present in sufficient concentrations (Figure A10), and the EAAI was calculated at 1.8, comparable to a whole egg (1.9) and higher than soybean meal (1.6). This indicates that duckweed could serve as a high-quality, plant-based protein source suitable for human diets.

#### 3.3.3. Duckweed Shows Promising Feed Value Considering Its Composition, Energy, and Digestibility Parameters

Table 8 presents the chemical composition, energy value (VEM, VEVI), and protein value (DVE, OEB) of dried duckweed harvested from S1 and S2 throughout the growing season (Mix 4). These parameters are specifically relevant for evaluating feed value in ruminants such as cattle. For context, values are compared to those reported in two Dutch duckweed studies [50,51], as well as soybean meal and grass silage, which are common components in ruminant feed.

As discussed in Section 3.1.1, systems 1 and 2 showed a stable and high protein production, which is reflected in the crude protein values of Mixes 1–4. The average protein content of Mix 4 (373 g/kg DM) is comparable to values reported in previous duckweed studies (396–268 g/kg DM), and significantly higher than grass silage (142 g/kg DM), although lower than soybean meal (526 g/kg DM).

Duckweed also showed a notably high ash content (177 g/kg DM), considerably above that of soybean meal (71 g/kg DM). Interestingly, the ash content in duckweed appears not to be strongly influenced by nutrient availability in the water body [56]. Pagliuso et al. observed that duckweed grown in low-nutrient waters tends to exhibit higher ash and fibre content with lower protein, whereas nutrient-rich conditions (such as in this study) result in higher protein and ash but lower fibre. Duckweed’s ash content has been reported to vary widely, from 7% to 36% DM [57]. While high ash content may not severely impact ruminant digestion, it could negatively affect digestibility in monogastric animals [43,44,45].

The Neutral Detergent Fibre (NDF) content of the duckweed was 240 g/kg DM, lower than grass silage but higher than soybean meal. This aligns with expectations for duckweed grown in high-nutrient media and is comparable to values found in the literature using a comparable medium [51].

The net energy value for ruminants was evaluated and expressed in the Feed Unit Lactation (VEM) and in the Feed Unit Growth (VEVI). Compared to soybean meal, Mix 4 had lower VEVI and VEM values, reducing its economic value as a high-energy protein feedstock. However, compared to the other duckweed studies and grass silage, duckweed cultivated in systems 1 and 2 showed better energy values. Moreover, the DVE value, representing the amount of true protein absorbed in the small intestine, was 202 g/kg DM in Mix 4. This was lower than soybean meal, but higher than grass silage and what was reported in other duckweed studies [50,51]. This indicates that duckweed offers intermediate protein value for ruminants.

The OEB value, which reflects the balance on rumen level between potential microbial protein synthesis based on degradable protein and microbial protein synthesis based on fermentable organic matter, was also measured. All feedstocks in Table 8 had a positive OEB, meaning that there was more rumen-degradable protein in the feed than energy available for microbial growth, leading to excess protein being converted to ammonia, which is then excreted. Compared to soybean meal, Mix 4 had a significantly lower OEB (83 vs. 217 g/kg DM).

In the research by Holshof et al., the energy and protein value of duckweed was considered overestimated due to its low organic matter digestibility (OMD). Using the Tilley and Terry method, they measured an in vitro OMDcof of only 63.4%, which they considered low compared to fresh grass (80%) [30,50]. In the present study, the digestibility of organic matter was also assessed in vitro on all four mixes. However, due to suspected microbial death in the inoculum (rumen fluid from sheep) over time, an alternative cellulase-based method was used to measure enzymatic degradation. For Mix 4, the cellulase-based digestibility reached 95.4%, whereas the rumen fluid-based digestibility reached 66.4%. Based on the cellulase digestibility and ash content, the OMDcof of 83.5% was calculated, which is significantly higher than the value reported by Holshof et al. [50]. The discrepancy may result from differences in methodology. A true in vivo digestibility trial would be required to resolve this variation and confirm the real digestibility potential of duckweed. This was tested in the research by Kroes et al., where an in vivo organic matter digestibility of 73.4% was reported for duckweed growing on water fertilised with agricultural waste streams [51]. Additionally, the variation in OMDcof and enzymatic digestibility between the different duckweed harvests (Mixes 1–3) was minimal, suggesting consistency over time.

The overall feed value of dried duckweed from S1 and S2 indicates strong potential as a complementary protein source in ruminant diets. Its composition, energy value, and protein value suggest that duckweed fits between grass silage and soybean meal in terms of nutritional quality. For monogastric animals, the high ash and fibre content may limit inclusion rates, but the crude protein concentration remains promising. Duckweed could serve as a functional feed ingredient in tailored monogastric diets, provided that formulation takes into account its mineral content and digestibility. Further targeted in vivo trials are recommended to fully assess its applicability in (non-)ruminant feeding strategies.

## 4. Conclusions

This study confirms the potential of duckweed-based treatment systems to simultaneously recover nutrients from pig manure processing effluents and produce protein-rich biomass under temperate climate conditions. Two duckweed systems supplemented with diluted liquid fraction (LF) and nitrification–denitrification effluent (NDNE) achieved consistent growth, yielding 8 tonnes of dry matter/ha/year and 3 tonnes of protein/ha/year. Nutrient removal rates (1.2 g N/m^2^/day and 0.12 g P/m^2^/day) were consistent with the literature and comparable to other biological treatment systems in terms of N and P. Cultivation on constructed wetland effluent alone was not successful, primarily due to salinity stress (EC up to 12 mS/cm). Even in the more successful systems, duckweed growth ceased after ~100 days despite suitable weather and non-toxic single-nutrient concentrations. The combination of increasing pH, potentially raising ammonia toxicity, and a gradual rise in salinity likely induced progressive salt stress, contributing to early growth cessation. These findings highlight the importance of dimensioning duckweed systems based on the most limiting nutrient, whose accumulation may impair performance. Physical adjustments, such as increasing surface area or pond depth, can enhance nutrient removal and buffering capacity. From a feed perspective, harvested duckweed showed strong nutritional potential. It provided a complete amino acid profile suitable for broiler diets. It is also a promising feed ingredient for ruminant rations due to its relatively high energy value (VEM, VEVI) and high intestinal digestible protein content (DVE). For monogastric species, the relatively high ash and fibre content may limit inclusion rates, although duckweed could still be incorporated in lower proportions as a micronutrient source or in processed form. While not a full substitute for high-value feedstocks such as soybean meal, duckweed could represent a valuable, locally produced, and circular protein source for diversified feed applications.

## Figures and Tables

**Figure 1 plants-14-02680-f001:**
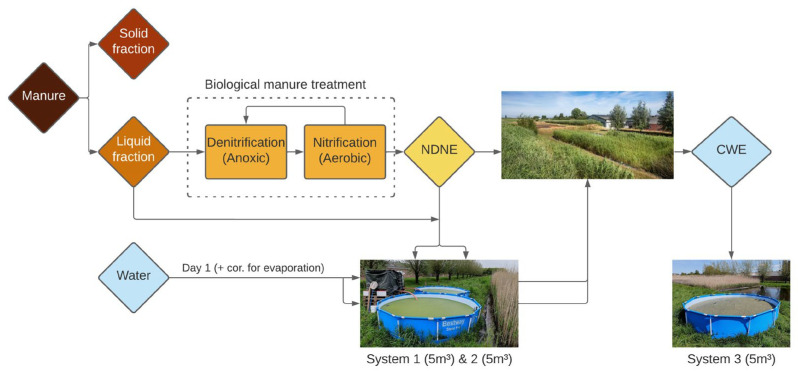
Schematic overview of the pig manure treatment process (with NDNE = nitrification–denitrification effluent and CWE = constructed wetland effluent = dischargeable water) at the pig farm and the composition of the growing medium of the three duckweed treatment systems.

**Figure 2 plants-14-02680-f002:**
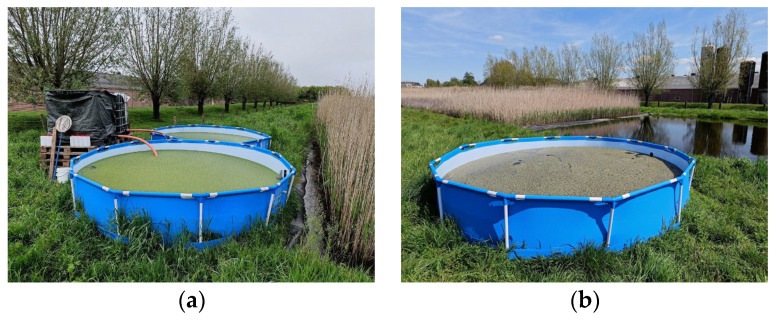
Picture of (**a**) S1 and S2 filled with a mixture of water, nitrification–denitrification effluent (NDNE), and liquid fraction of pig manure (LF), in which duckweed is growing. The black plastic-covered IBC containers storing the fertilizer mix (10% LF and 90% NDNE) can be seen next to them; (**b**) Picture of S3, filled with effluent from the wetland in which duckweed is growing.

**Figure 3 plants-14-02680-f003:**
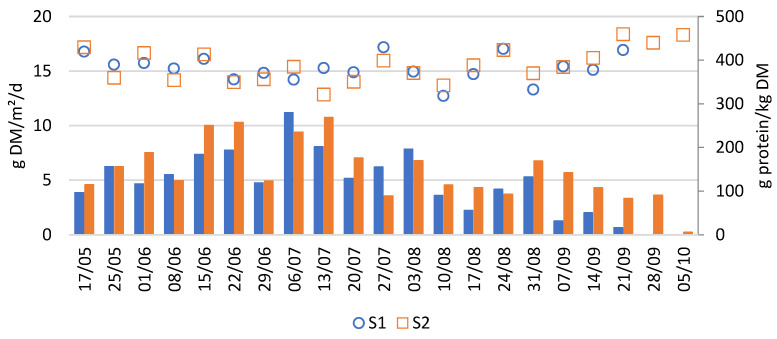
Obtained productivity (bars) and protein concentration (dots and squares) of duckweed grown in a pig manure-based medium in two identical systems (S1 in blue and S2 in orange). Productivity is presented in g duckweed dry matter per m^2^ system surface per day and was calculated with the amount of duckweed harvested on a given day, indicated on the *x*-axis, divided by the number of days since the previous harvest.

**Figure 4 plants-14-02680-f004:**
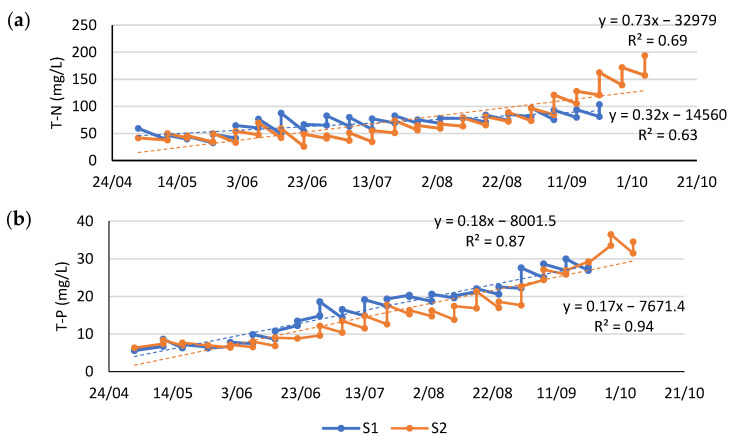
(**a**) Nitrogen and (**b**) phosphorus concentration of the growing medium in systems 1 and 2 measured of the growing season (*x*-axis presents the data of sampling as dd/mm). Samples were taken before and after fertilising every week while the surface of the system was covered with duckweed.

**Figure 5 plants-14-02680-f005:**
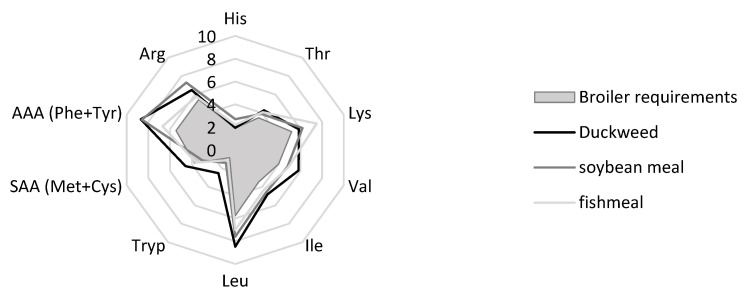
Representation of the essential amino acid composition of a duckweed protein (Mix 4) as a percentage of the total crude protein. For comparison, the broiler requirements and the EAAs% of two alternative protein sources used in broiler feed are presented [21,48,52].

**Table 1 plants-14-02680-t001:** Total N, total P, and N to P ratios of LF (liquid fraction) and NDNE (nitrification–denitrification effluent), together with the calculated composition of the mixture after a non-linear solver technique maximising the NDNE fraction within the present restrictions. The resulting mass fraction gives the composition of the start medium of S1 and S2.

	Total N (mg/L)	Total P (mg/L)	N:P Ratio	Mass Fraction (%)
LF	2741	178	15.4	1.48
NDNE	361	73.3	4.92	11.4
Water	0	0	0	87.1
Mixture	81.7	11	7.4	100
Restrictions	<350	<11	=7.4	

**Table 2 plants-14-02680-t002:** Summary of the analysed parameters to assess the nutritional value of duckweed as a feed ingredient, including the unit and referring to the analytical method or calculation method.

Parameter	Unit	Method/Calculation
Moisture content	g/kg	152/2009/EC [23]
Crude protein (CP)	g/kg DM	Kjeldahl, ISO 5983-2 [24]
Crude ash (CA)	g/kg DM	ISO 5984 [25]
Neutral Detergent Fibre (NDF)	g/kg DM	[26]
Crude fat (CF)	g/kg DM	Soxhlet, ISO 6492(A) [27]
Sugars (SUG)	g/kg DM	Luff Schoorl (152/2009/EC) [23]
Starch (ST)	g/kg DM	Amyloglucosidase (NEN3574) [28]
Cellulase digestibility of organic matter (OMDc)	%	[29]
Organic matter digestibility coefficient (OMDcof.)	%	[30]
Dry matter (DM)	g/kg	=1000 − moisture content
Organic matter (OM)	g/kg DM	=DM − CA
Non-starch polysaccharides (NSP)	g/kg DM	=OM − CP − ST − CF − SUG
Remaining non-starch polysaccharides (RNSP)	g/kg DM	=OM − CP − ST − CF − SUG − NDF
VEVI (Feed Unit Growth)	/kg DM	Based on Van Es [31]
VEM (Feed Unit Lactation)	/kg DM	Based on Van Es [31]
DVE (Digestible Crude Protein in the Small Intestine)	g/kg DM	Based on Tamminga et al. [32]
OEB (Degraded Protein Balance)	g/kg DM	Based on Tamminga et al. [32]

**Table 3 plants-14-02680-t003:** Overview of the cultivation period, the biomass and protein productivity of systems 1, 2, and 3 presented for fresh and dried duckweed.

	Unit	System 1	System 2	System 3	Previous Research [3]
Cultivation period	days	142	156	84 ^b^	175 ^a^
Total biomass production	kg fresh	110	138	4.89	
	kg DM	6.97	8.77	0.47	
Mean protein content	(%) of DM	38.0 ± 3.0	38.7 ± 3.9	14.8 ± 3.8	33.0 ± 3.6
Dry biomass production	ton/ha/year	6.85	8.61	0.46	10.7
Protein production	ton/ha/year	2.61	3.33	0.07	3.4

^a^ Assumption made to calculate the yearly dry biomass and protein productivity; ^b^ Biomass samples were taken between that period but there was no continuous growth, duckweed died after 24/8.

**Table 4 plants-14-02680-t004:** Average overall system removals and the specific removal by weekly harvesting the duckweed from S1 and S2, together with their standard deviation, measured from the first till the last harvest.

	System Removal			Specific Removal by Harvest					
	mg/m^2^/d			mg/m^2^/d			% from Overall System Removal		
T-N	1198	±	744	311	±	145	48	±	74
T-P	125	±	150	72	±	40	97	±	136
PO_4_	88	±	498						
SO_4_^2−^	577	±	6379						
T-S	535	±	934	40	±	20	26	±	84
Cl	2816	±	7718						
K	4652	±	9547	255	±	144	8	±	14
Ca	267	±	1034	98	±	80	59	±	124
Mg	75	±	134	21	±	11	38	±	44
Na	1509	±	3096	50	±	24	4	±	6
Fe	120	±	164	8	±	9	11	±	23
Mn	4	±	11	2	±	2	50	±	81
Cu	9	±	11	0.2	±	0.1	4	±	8
Zn	21	±	23	0.9	±	0.9	9	±	20
Co	0.07	±	0.21	0.004	±	0.003	7.1	±	11
BOD	700	±	1800						
COD	9219	±	18,664						
TSS	17,208	±	17,161						

**Table 5 plants-14-02680-t005:** Minimal and maximal measured concentrations measured in systems 1 and 2, together with the optimal growing ranges for duckweed and the maximum element concentration before reaching toxicity.

	Min	Max	Optimal	Maximal	Unit	Evaluation
pH	7.41	8.60	6.5–7.5 ^α^	5.0–9.0 ^α^		Sub-optimal
EC	2.47	9.98	0.6–1.4 ^α^	<10.9 ^α^	mS/cm	Sub-optimal
T-N	25.8	194	2.8–350 ^α^	<2100 ^α^	mg/L	Optimal
T-P	5.58	36.5	0.4–11 ^α^	<55 ^α^	mg/L	Sub-optimal
PO_4_^3−^	8.25	86.3	0.4–11 ^α^	<55 ^α^	mg/L	Detrimental
SO_4_^2−^	110	674	48–1900 ^α^	<4800 ^α^	mg/L	Optimal
T-S	58.7	123			mg/L	
Cl^−^	213	1013	0.4–36 ^α^	<3500 ^α^	mg/L	Sub-optimal
K	311	1657	39–780 ^α^	<2000 ^α^	mg/L	Sub-optimal
Ca	32.1	76.1	20–400 ^α^	<2000 ^α^	mg/L	Optimal
Mg	10.6	25.8	5.0–97 ^α^	<1200 ^α^	mg/L	Optimal
Na	120	499	120–230 ^α^	<3400 ^α^	mg/L	Sub-optimal
Fe	3.30	12.9	<27.9 ^β^	<100 ^β^	mg/L	Optimal
Mn	0.06	0.62	<54.9 ^β^	<274.5 ^β^	mg/L	Optimal
Cu	0.16	0.71	<3.2 ^β^	<6.3 ^β^	mg/L	Optimal
Zn	0.19	1.34	<6.5 ^β^	<65.3 ^β^	mg/L	Optimal
Pb	<LOD	0.27			mg/L	
Co	0.004	0.02			mg/L	
Cd	<LOD	0.01			mg/L	
Cr	<LOD	0.04			mg/L	
Ni	<LOD	0.12			mg/L	

Sources: ^α^ [9]; ^β^ [41].

**Table 6 plants-14-02680-t006:** Minimum and maximum nutrient concentrations in duckweed compared to soybean meal (both 100% DM) [48] and feedstuff (or feed ingredient) requirements for broilers, pigs, and cattle. Data sources: a [43]; b [46]; c [47]; d [44]; e [45].

	Duckweed		Soybean Meal	Feedstuff Requirements			
	Min	Max	Mean	Broilers	Pig *	Cattle **	Unit
T-P	10	19	7		4 ^a^		g/kg
T-S	7	9	5			1.5 ^e^	g/kg
Ca	6	36	3.9	8–10 ^d^	4.5 ^a^		g/kg
Mg	3	5	3.2	0.6 ^d^	0.4 ^a^	1–2 ^e^	g/kg
Na	5	15	0.13	1.2–2 ^d^	1 ^a^		g/kg
K	26	63	24.3	3 ^d^	1.7 ^a^	6–7 ^e^	g/kg
Fe	260	4136	201	0.08 ^d^	0.04 ^a^	0.05 ^e^	mg/kg
Mn	174	877	44	0.06 ^d^	0.002 ^a^	0.02–0.04 ^e^	mg/kg
Zn	68	825	57	0.07–0.13 ^c^	0.05 ^a^–0.15 ^c^	0.018–0.06 ^c^	mg/kg
Cu	15	76	17	8 ^d^–25 ^b^	3 ^a^–25 ^b^	30–35 ^b^	mg/kg
Co	0.4	1.6	0.3				mg/kg

* for fattening finisher pigs (80–120 kg); ** for growing and finishing cattle (200–450 kg).

**Table 7 plants-14-02680-t007:** Non-essential heavy metals and metalloids, considered toxic in low quantities, measured in dried duckweed samples at the beginning (Mix 1), mid (Mix 2) and end (Mix 3) of the growing season (see Table A1) together with their max allowable feed concentrations according to the EU Directive 2002/32/EC [49].

	Mix 1	Mix 2	Mix 3	Max *	Unit
Cd	0.08	0.02	0.03	1	mg/kg dried duckweed
Pb	0.11	0.13	0.13	5	mg/kg dried duckweed
As	0.42	0.23	0.18	2	mg/kg dried duckweed
Hg	<LOQ **	<LOQ **	<LOQ **	0.1	mg/kg dried duckweed

* Maximum allowable concentrations represent the strictest limits as defined in EU Directive 2002/32/EC, excluding any specific derogations or exemptions [49]; ** Limit of quantification (0.01 mg/kg).

**Table 8 plants-14-02680-t008:** Chemical composition, OM digestibility, net energy value for dairy cattle (VEM), fattening cattle (VEVI) and protein value for cattle (DVE, OEB) of 4 dried duckweed samples with a composition as described in Table A1, compared to duckweed values obtained from the literature [50,51]—soybean meal [55]—grass silage [33]; abbreviations are explained in Materials and Methods, Section Proximate Analysis.

	Mix 1	Mix 2	Mix 3	Mix 4	Duckweed		Soybean Meal [55]	Grass Silage [33]	Unit
					[51] ^a^	[50] ^b^			
Dry matter	921	920	924	917	48 *	886	880	297	g/kg
Crude protein	373	354	391	373	396	268	526	142	g/kg DM
Crude fat	25	22	24	25	44	28.9	18	44	g/kg DM
Ash	176	192	169	177	164	173	71	167	g/kg DM
Sugar				7	6		92		g/kg DM
Starch				58			19		g/kg DM
NSP				360					g/kg DM
RNSP				120	174 **				g/kg DM
NDF				240	216		142	474	g/kg DM
OMDc	93	93	92	95	83.2				%
OMDcof	81.4	80.9	80.7	83.5		63.4			%
VEM	902	869	896	930	839	826 ***	1178	829	/kg DM
VEVI	956	916	947	994	860		1275	854	/kg DM
DVE				202	130	118 ***	257	47	g/kg DM
OEB				83	214	85 ***	217	25	g/kg DM

* Dry matter on fresh duckweed, other values on dried duckweed; ** under the assumption that there was no starch in the analysed duckweed sample; *** calculated based on regression formulas of grass pellets, assuming the same digestibility counts for dried duckweed; ^a^ duckweed grown on water fertilised with agricultural waste streams; ^b^ duckweed grown on natural waterways.

## Data Availability

The data presented in this study are available on request from the corresponding author.

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
