# Peer review of "A Pilot-Scale Evaluation of Duckweed Cultivation for Pig Manure Treatment and Feed Production"

_plants, 2025, doi:10.3390/plants14172680_

Round 1
Reviewer 1 Report
Comments and Suggestions for Authors
Paper by Lambert et al, entitled “A pilot-scale evaluation of duckweed cultivation for pig manure treatment and feed production”.
This is an extensive, well detailed paper that contains a large amount of important information. Overall, the paper is well written, but I would like to suggest some minor amendments to improve clarity.
In the abstract it is stated that 8 tonnes of dry matter contains 2.8 tonnes of protein (lines 27/28). A little bit down (line 31) a slightly different number, 373 g/kg is mentioned. Why the small difference, and is it really necessary to give twice the same info?
Line 43/44 refers to “pig manure wastewater” without explaining the meaning of this term. Please explain.
Is it correct that unlike this project, the Spanish group used “pig manure” (line 53) including solids?
Line 73, “Devlamynck et al”, please insert citation number.
The introduction sets out the context of the study very nicely, but what is lacking is a clear aim.
Figure 1, what is “Disch. Water”? In general, legends to figures and tables can be improved by explaining used abbreviations and terms.
Line 130. A nutrient mix stored in an IBC at room temperature will gradually change due to microbial activity. Was the mix re-analysed prior to use with the duckweed?
Line 142-145. Nutrient replacement volumes don’t seem to consider “other” forms of nutrient removal (ammonification, denitrification/nitrification, algal competition and so on.
Line 147, what is the effluent this refers to? Effluent from solid/liquid slurry separation, or effluent at end of wetland? Same applies to the legend of figure A3.
My only more major comment relates to the lack of quantification of duckweed biomass/surface cover at the start of each interval. I understand that this precludes calculation of RGR, but the reader has at present no idea at all what is going on. How much surface cover is there? Is surface cover low enough to encourage algal growth, or high enough to reduce duckweed growth? What does it mean that a “part of the duckweed is harvested” (line 168/168) What part? Is this undermining all biomass quantification? In figure 3 amount of harvested duckweed is shown; so what determined how much duckweed (a part …?) was harvested. See also lines 396-398.
Related to the above question. It is not unusual to calculate N and P uptake as mg per m2 duckweed per day. I guess that since duckweed cover is not quantified, this has been calculated as m2 water surface? This needs to be clearly explained.
Some of the formula’s don’t explain all abbreviations. For examples, line 346-347; what is V or c? Line 351-352 uses c in a different way, as plant content, and not concentration.
Figure 3. Please explain what are the two types of bars.
Table 4. As I understand the term “total removal”, some these nutrients might not be removed at all but reside in the periphyton/biofilm or in the sediment. Is this term misleading?
Line 497-498. I understand the argument used in these lines, but this argument hinges on how the N and P analysis were done, specifically if samples were filtered it would have removed sediment-contained Nand P. Perhaps this can be clarified in the relevant section of the M&M?
Line 564. The imbalance between N and P additions and removals is discussed in these lines. I am surprised that the possibility that predicted N and P uptake rates were overestimates, is not discussed.
Figure 4, and several other figures, have a strange symbol on the X-axes.
Line 631-632 to some extent repeats line 402-404 where it relates to growth. I recommend removing repetition.
Lines 670-678 seems out of context in this section and can perhaps be removed?
Lines 693-700. The weekly monitoring data are very interesting, especially given that temporal variations are found. Why not show these data? I can’t find them in the supplemental info either?
Line 741; what is meant by “not the digestible fraction”?
Section 3.3.4. The issue of microbial safety is important. However, I feel that section 3.3.4 does no justice to this important topic. Reliance on single and pooled samples, some of which would have been stored for considerable time as dried biomass, makes this section very weak. I recommend to remove this part of the study.
Some of the labels (months) in figures A7, A8 and A11 are in Dutch(?)
Table A8 and other places. Is there a reason to sometimes give concentrations in mg/kg and in other cases mg/L (compare tables A8 and A9 plus other cases).
Figure A11. My understanding is that NH3-N is toxic at substantially lower concentrations than 8mg/L
Reviewer 2 Report
Comments and Suggestions for Authors
This manuscript is a pilot plant study of the use of duckweed in pig manure treatment and the potential use of this plant in animal feed. The study is interesting and may be of practical use, considering the results obtained. The manuscript is well-written, but some important considerations should be taken into account:
1) Abstract: Do not use abbreviations, or they must be previously explained.
2) Lines 137, 144: What is BE?
3) Lines 139-140: The addition of N and P is based on the same amount that must be removed from the system; however, this is one of the variables to be determined. This part is a bit confusing and needs to be explained more clearly. Given the experimental design, the source of variation in the results can be large, as can be seen from the results obtained.
4) Line 141: Have these results been prepared under the same culture conditions as the experiments in this work?
5) Overall, it seems that the experimental design is based on several assumptions. Explain or discuss this fact because, indeed, this is a very important source of variation.
6) S3 is a single pool, can the results from this pool be considered significant?
7) Lines 186-188: What is the difference between dry weight and dry matter?
8) Sample Preparation: What is the rationale for considering this type of sampling somewhat unusual?
The Mix 2 sample had one less week of sampling; was this taken into account? Given this, the sampling performed was irregular.
9) Equations 2 and 3, if the results are per week, the units should be mg/m2/w. Or was it divided by 7 days? Explain.
10) Figure 3: Explain in Materials and Methods how the productivity sample was collected. Was part of the sample left behind? Was it sampled by m2?
11) Lines 465-472: Frankly, it's not possible to draw any conclusions from these data. These deviations exceed what can be considered random variation. As I indicated in a previous comment, I believe the experimental design is based on too many assumptions and has too many sources of variation. Relying on the results, with this in mind, is very risky.
12) Lines 480-489: The authors mix N and TKjN in this discussion. Please explain in more detail based on your results.
13) Figure 4 (and figures in the Annex): Is the X-axis correct? What do the X-axis symbols mean? This figure doesn't look like it's designed correctly.
14) Table 5: T-P is total phosphorus. This determination is not included in Materials and Methods. But considering the data in this table, how is it possible that there is more PO43- than T-P?
15) The abbreviation for liter is conveniently capitalized, L, mL. Check it out.
16) Microbiological analysis: It is not adequate to draw conclusions from a single sample, however, it is strange that, given its origin, the E. coli and enterobacteria count is so low.
17) Figure A5 could be included in the body of the manuscript.
18) Table A9: pH units can be variation pH/d.
19) References: Species names should be italicized and spelled correctly. Respect scientific nomenclature.
Round 2
Reviewer 2 Report
Comments and Suggestions for Authors
Dear authors, the responses to my comments on the manuscript appear to be correct, however, they are not reflected in this version of the manuscript. The manuscript appears similar to the previous version. Please highlight in yellow all changes made to the previous version and verify that all required changes have been included in the new version of the manuscript.
Round 3
Reviewer 2 Report
Comments and Suggestions for Authors
The changes were satisfactory. The manuscript can be accepted for publication.